# Evaluation of Selected Plant Essential Oils for Aphid Pest Control in Integrated Pest Management

**DOI:** 10.3390/insects16040353

**Published:** 2025-03-27

**Authors:** José Luis Casas, María López Santos-Olmo, Aitor Sagarduy-Cabrera, Mᵃ Ángeles Marcos-García

**Affiliations:** 1Unidad Asociada de I+D+i al CSIC “Interrelaciones Insecto-Patógeno-Planta y sus Agentes de Biocontrol” (IPAB), Research Institute CIBIO (Centro Iberoamericano de la Biodiversidad), Scientific Park, University of Alicante, Ctra. San Vicente del Raspeig s/n, 03690 San Vicente del Raspeig, Alicante, Spain; marcos@ua.es; 2Research Institute CIBIO (Centro Iberoamericano de la Biodiversidad), Scientific Park, University of Alicante, Ctra. San Vicente del Raspeig s/n, 03690 San Vicente del Raspeig, Alicante, Spain; maria.lopezs@gcloud.ua.es (M.L.S.-O.); aitorsc98@gmail.com (A.S.-C.)

**Keywords:** *Myzus persicae*, biological control, *Sphaerophoria rueppellii*, cypress, laurel, rosemary, integrated control, sustainable agriculture

## Abstract

Pesticide residues are a major contributor to biodiversity loss in both agricultural and natural ecosystems. In addition to the adverse health effects on farmers and consumers resulting from residue exposure, the high capacity of pests, such as aphids—responsible for severe economic losses in greenhouse crops—to develop resistance to pesticides has led to pest resurgence and significant economic damage. This underscores the urgent need for natural alternatives to chemical insecticides to ensure both plant protection and agricultural productivity. Effective aphid management relies on Integrated Pest Control, which combines the use of authorised chemical products with biological control by natural enemies. This study aims to optimise this dual approach by evaluating three essential oils for aphid control while ensuring their compatibility with natural enemies. The essential oils tested—extracted from rosemary, laurel, and cypress—demonstrated high aphid mortality (>80%) while causing minimal adverse effects on one of their natural predators, the hoverfly (*Sphaerophoria rueppellii*). These findings represent a promising step towards the widespread application of plant-derived essential oils as biopesticides.

## 1. Introduction

Some aphid species (Hemiptera: Aphididae) are regarded as significant pests affecting a wide variety of agricultural crops in temperate climates [1]. Global crop losses attributed to aphid infestations are estimated to amount to tens of millions to billions of US dollars annually [2]. Among the most damaging species is the green peach aphid *Myzus persicae* (Sulzer, 1776), a cosmopolitan and highly polyphagous pest [3] with an exceptionally broad host range, encompassing more than 400 species across 50 plant families [4,5], predominantly in greenhouse crops. This species has developed resistance to a wide range of commonly used chemical pesticides [6,7,8], posing significant challenges to effective pest management.

*M. persicae* is a phloem-sap-feeding insect that causes direct damage by sucking the sap [9] and covering plant tissues with honeydew, which significantly disrupts photosynthesis and alters plant hormonal balance, among other effects [10]. Moreover, this aphid is capable of transmitting more than 100 plant viruses, both persistent and non-persistent, to numerous species of agricultural, forestry, and ornamental interest [2,11]. Its rapid population growth can lead to infestation levels that are extremely difficult to control, highlighting the urgent need for sustainable solutions that ensure plant protection and productivity while remaining compatible with the use of natural enemies [12].

Within integrated pest control, a fundamental approach is the use of natural enemies, primarily predators, and parasitoids to maintain pest populations below the economic damage threshold while ensuring environmental sustainability and safeguarding the health of animals, farmers, and consumers. However, when preventive measures have not been implemented in aphid management, control must rely on an integrated strategy that combines authorised chemical products with natural enemies [13]. The synergistic interaction between two or more control techniques, whereby their combined effect reduces pest populations more than would be expected from their individual effects, is known as superadditivity [14]. This study proposed optimising this dual-control strategy by identifying the most effective plant essential oil and the most appropriate doses for controlling the green peach aphid, while ensuring minimal impact on one of its key natural enemies.

One of the primary native predators of aphids is the hoverfly *Sphaerophoria rueppellii* (Wiedemann, 1830) (Diptera: Syrphidae), a species naturally found within greenhouses and capable of withstanding the extreme humidity and temperature conditions present in such environments [15]. Unlike other natural enemies, which may be less effective under these conditions, *S*. *rueppellii* has been recognised as one of the most efficient aphid predators in greenhouse crops [16,17,18]. The predatory larvae of this syrphid feed on various soft-bodied insect pests, including thrips, psyllids, mealybugs, and lepidoptera and acari species, with aphids being their main prey [19]. Meanwhile, adults contribute significantly to pollination, playing a dual role in integrated pest management [16,20]. Under field conditions, *S. rueppellii* has been shown to reduce aphid populations by 84%, thereby enhancing crop yields [21,22]. Furthermore, this species has demonstrated resistance to organophosphate insecticides [21]. However, certain compounds, such as the neonicotinoids imidacloprid and thiamethoxam, are toxic to *S. rueppellii* [17,23]. Some synthetic insecticides, such as flonicamide, are effective against *M. persicae* but prove lethal to *S. rueppellii*, as syrphids are more sensitive to insecticides than parasitoids due to their higher feeding rates and lower detoxification capacity [17,23]. Consequently, the integrated control of *M. persicae* necessitates chemical control strategies that incorporate natural products compatible with the presence of natural enemies [24].

A considerable number of plant species have been shown to possess properties that repel, kill, or inhibit the growth of harmful organisms [25]. These effects are attributed to the presence of essential oils (EOs), whose use may be of significant interest in integrated pest management strategies for aphid control [26]. Consequently, EOs are proposed as a viable and more environmentally friendly alternative not only for insect control but also for weed management [27,28]. EOs are highly volatile, low-molecular-weight organic compounds, with monoterpenes and sesquiterpenes as the predominant groups, that have demonstrated efficacy against various pests, including aphids [26]. It has been demonstrated that species within the Lamiaceae family contain compounds with insecticidal properties [29], while some Apiaceae species exhibit toxicity against *M. persicae* [30]. Certain natural sesquiterpenoids, such as geranylacetone and nerylacetone, exhibit deterrent activity against *M*. *persicae* as these compounds and their epoxy derivatives inhibit aphid settlement on treated leaves [31]. Additionally, other compounds present in rosemary and laurel have also been shown to deter this aphid species [32]. Aphid repellence has also been demonstrated not only as an effect of specific essential oils [33] but also of individual components such as of citral and linalool [34]. Another advantage of EOs is that certain natural enemies and pollinators, such as hoverflies, show tolerance to them [35]. As a result, these compounds are considered safe for environmental conservation, ecosystem functioning, and human health [36,37], making them suitable for use as biopesticides against aphids [26].

The present study aims to chemically characterise and assess the insecticidal efficacy of three EO derived from taxonomically unrelated plant families—rosemary (*Salvia rosmarinus* Spenn., Lamiaceae), laurel (*Laurus nobilis* L., Lauraceae) and cypress (*Cupressus sempervirens* L., Cupressaceae)—for their potential use in integrated pest management programs against aphids. Additionally, the effects of these EO on one of the main natural aphid predators, the larvae of the hoverfly *S. rueppellii* (Wiedemann, 1820), will be evaluated. This approach seeks to demonstrate the potential additivity of two control measures that may be effective independently but even more so in combination. Lastly, correlating the chemical composition of these EO with their toxicity to aphids could shed light on the basis of the biological activity of these compounds as biopesticides.

## 2. Materials and Methods

### 2.1. Plant Material

Rosemary (*Salvia rosmarinus* Spenn.), laurel (*Laurus nobilis* L.), and cypress (*Cupressus sempervirens* L.) were collected from the Biological Station–Botanical Garden of the University of Alicante (Ibi, Alicante, Spain: 38°29′36″ N, 0°39′43″ W). The aerial parts were dried for 3–5 days at room temperature (19–22 °C) in the laboratory. Once dried, the leaves were separated from the remaining plant material and ground using an electric grinder.

### 2.2. Extraction of EOs

EOs were extracted from 50 g dw plant samples by hydrodistillation using a Clevenger-type apparatus [38] for 3 h, with a ratio of 10 mL distilled water per gram of dried plant material [39]. The oil fraction was collected, dehydrated with anhydrous sodium sulphate, and stored in amber glass vials at 4 °C until use. The volume and weight of the EO fraction were also measured to calculate the oil extraction yield.

### 2.3. GC-MS Characterization of EOs

The extracted EOs were analysed by gas chromatography-mass spectrometry using an Agilent 7890A gas chromatograph coupled to an Agilent 5975C mass spectrometer (Santa Clara, CA, USA). The specific analytical conditions followed those described previously [39]. Component identification was performed by comparing the obtained mass spectrum with those stored in the NIST23 library. The relative abundance of each compound expressed as a percentage, was determined from peak areas without applying response factors. Identified compounds were classified into lipid families according to the Brite Hierarchy (KEGG) (https://www.genome.jp/kegg-bin/show_brite?br08002.keg, accessed on 15 November 2024) database for lipid-derived compounds and the Lotus Natural Products Online (https://lotus.naturalproducts.net/, accessed on 20 November 2024) for non-lipid compounds.

### 2.4. Insect Rearing

The green peach aphid *Myzus persicae* (Sulzer, 1776) was reared on pepper (*Capsicum annuum* L. var. Infante, Ramiro Arnedo, S.A., Calahorra, La Rioja, Spain) plants grown in trays containing a 1:2 vermiculite:peat mixture. The plants were kept inside mesh cages (28.5 × 60 cm) within growth chambers set at 24 ± 1 °C, 65% relative humidity, and a 14L:10D photoperiod. Second-instar larvae (L2) of the predatory hoverfly *Sphaerophoria rueppellii* (Diptera, Syrphidae) were supplied by BioNostrum Pest Control, S.A. (Alicante, Spain) and used immediately upon arrival.

### 2.5. Bioassays

The insecticidal activity of EOs was assessed using contact toxicity bioassays on both aphids and syrphids. All bioassays were conducted in plastic Petri dishes (5.6 cm in diameter), each containing two filter paper discs at the base to maintain adequate humidity within the dish.

#### 2.5.1. Contact Toxicity Bioassays Against Aphids

The insecticidal effect of EO on aphids was tested at three concentrations: 10, 4, and 2 μL/mL in acetone. A 15 mm pepper leaf disc was placed on the moistened filter paper at the base of each Petri dish, and ten apterous aphids were placed on the leaf disc. A total volume of 1 mL of each EO solution was prepared and uniformly sprayed over each dish. Control groups included aphids that were treated with acetone. Following treatment, the Petri dishes were maintained in a growth chamber set at 24 ± 1 °C, 65% RH and a 14L:10D photoperiod. Aphid mortality was recorded after 24 h.

#### 2.5.2. Contact Toxicity Bioassays Against Syrphids

The insecticidal activity of EO on syrphids was assessed at concentrations of 10 and 4 μL/mL in acetone. Two second-instar larvae of the hoverfly were placed directly onto the moistened filter paper in each Petri dish. After treatment, the Petri dishes were maintained under the same conditions described above, and mortality was recorded after 24 h. Control groups comprised larvae exposed solely to acetone (control 1) and untreated larvae maintained on moistened filter paper (control 2) to preclude the possibility that mortality was attributable to starvation. Syrphid larvae mortality was recorded after 24 h.

### 2.6. Statistical Analysis

Aphid mortality data are presented as the average of 10 Petri dishes, each containing ten aphids (n = 10), and expressed as the mean percentage ± standard deviation. Statistical significance (*p*) was analysed using Welch’s ANOVA, followed by Games-Howell as a post-hot test at *p* ≤ 0.05. Syrphid mortality data represent the average of 15 Petri dishes, each containing two syrphid larvae (n = 15). Statistical significance (*p*) was analysed using ANOVA, followed by Tukey HSD post-hoc test at *p* ≤ 0.05. Statistical analyses were conducted using SPSS (v. 29.0.1.0). Lethal concentration 50 (LC_50_) values were determined by Probit analysis [40] using a Probit analysis spreadsheet calculator [41].

## 3. Results

### 3.1. Essential Oil Characterization

The essential oils (EOs) extracted from three plant species—rosemary, laurel, and cypress—were characterised prior to evaluating their biopesticidal potential. Hidrodistillation yielded EO extraction rates of 1.18% ± 0.15, 0.73% ± 0.03, and 0.43% ± 0.03, respectively (Table 1). Gas chromatography-mass spectrometry (GC-MS) analysis identified more than 98% of the compounds present in the three EOs. These compounds were classified into their respective chemical families according to the Brite Hierarchy and Lotus Natural Products, with the results shown in Table 1. Rosemary EO exhibited the highest phytochemical diversity, containing seven compound families, while laurel and cypress EO comprised four and five families, respectively. Monoterpenes were the predominant class in all three EOs, particularly in rosemary, where they accounted for over 80% of the total compounds. Sesquiterpenes formed the second more abundant group in all EOs, displaying an inverse relationship with monoterpene content: cypress EO, which had the lowest monoterpene proportion (Table 1), contained the highest percentage of sesquiterpenes. Together, mono and sesquiterpenes constituted 94% of the compounds in rosemary EO, 84% in laurel, and 95% in cypress. Diterpenes were absent in laurel, present in trace amounts in rosemary, and more prominent in cypress EO. Notably, phenylpropanoids constituted 13.53% of the identified compounds in laurel EO, whereas their presence in the other two EOs was well below 1% (Table 1).

Rosemary EO contained 121–122 identified compounds, while laurel and cypress EO had 142–149 and 107–112 compounds, respectively (the full composition of each oil is detailed in Appendix A). Table 2 presents the ten more abundant compounds in each EO, which accounted for 71%, 68%, and 76% of the identified compounds in rosemary, laurel, and cypress EO, respectively.

In all three EO, the most abundant compound belongs to the monoterpenes class, with 1,8-cineole dominating in rosemary and laurel EOs, and cyclofenchene in cypress EO. Rosemary and laurel EOs are clearly dominated by monoterpenes, as nine of their ten most abundant compounds belong to this group (Table 1). In contrast, while monoterpenes are also predominant in cypress EO, four sesquiterpenes appear among its ten most abundant compounds.

### 3.2. Essential Oils as Insecticides Against Aphids

In order to assess the biopesticide potential of the extracted EO against aphids, contact toxicity bioassays were performed. The results (Table 3 and Table 4) indicate that all three EOs achieved an aphid mortality rate exceeding 80% at the highest dose applied. A statistically significant difference was detected between groups by Welch’s ANOVA (*F* (9,36.142) = 58.384, *p* < 0.001). A Games-Howell posthoc test revealed that all three EOs led to significantly higher aphid mortality compared to the control, with the exception of laurel EO at 2 μL/mL. When comparing the efficacy of each EO at a given concentration, no significant differences were observed except at the 4 μL/mL dose, where laurel and cypress oils were significantly more effective than rosemary. Notably, cypress EO exhibited the lowest LD_50_ value (1.93 μL/mL), suggesting high potency at lower doses (Table 3).

### 3.3. Essential Oils Against Aphid Natural Enemies

The biological activity of the selected EOs was also tested against one of the main natural enemies of aphids, the syrphid larvae of *Sphaerophoria rueppellii*. Contact toxicity bioassays were conducted using the two highest concentrations of each EO, and the results are presented in Table 5. Regardless of the EO used, the average mortality rate in syrphids did not exceed 20%, and no statistically significant differences were detected (*F* (9,90) = 0.871, *p* = 0.554) among EOs or in comparison to the controls.

## 4. Discussion

The need for viable alternatives to synthetic insecticides is now widely recognised. Persistent organic pesticides pose a serious threat to the environment [42,43], yet effective control of insect pest populations in crops remains essential. Consequently, the development of effective and environmentally sound pest management techniques is a priority. However, any chemical control measures implemented must be compatible with other control strategies, such as biological control utilising natural enemies and predators, thus adhering to the principles of sustainable agriculture. In this context, our group is engaged in the search for EOs that are highly effective against aphids but only minimally harmful to their natural enemies. The efficacy of three Mediterranean aromatic plant EOs—*Thymus vulgaris*, *Rosmarinus officinalis* var. ‘prostratus’, and *Lavandula dentata*—as aphid biopesticides was recently reported [39]. In the present study, the focus was placed on the EO of three further species: rosemary (now renamed as *Salvia rosmarinus* [44]), albeit a variety of normal growth habits rather than the creeping ‘prostratus’ variety, laurel, and cypress. All three EOs showed highly promising characteristics as potential insecticides based on laboratory tests, with aphid mortality rates exceeding 80% at a dose of 10 μL/mL (Table 3) and only slight toxicity observed against the aphid natural enemy, *S. rueppellii* (Table 4). The similar insecticidal potential exhibited by these three species, despite belonging to different families (rosemary: Lamiaceae, laurel: Lauraceae, and cypress: Cupressaceae), prompts investigation into shared characteristics within their EOs.

Rosemary EO was primarily characterised by the monoterpenes camphor, 1,8-cineole, and cyclofenchene as the major compounds, comprising 41.1% of the EO’s composition (Table 2). This composition differed from that of the previously reported *S*. *rosmarinus* var. ‘prostratus,’ in which α-pinene, verbenone, and 1,8-cineole were the predominant compounds, accounting for 40% of the total [39]. Intriguingly, this variation in EO composition correlated with differences in aphid mortality. The rosemary EO used in this study demonstrated substantially higher efficacy (85% ± 21.7) than that reported for rosemary var ‘prostratus’ (41.7% ± 25.3) [39] at the same dosage. Ainane et al. [45] also identified camphor (31.16%) as the most abundant compound in rosemary harvested from various Moroccan locations, followed by β-caryophyllene (18.55%) and (Z, Z-)-3,4-dimethyl-2,4-hexadiene (9.08%). They reported high insecticidal activity of rosemary EO against *Tribolium confusum*, a stored-product pest, observing complete mortality at 0.035 μL/mL. Similarly, a rosemary EO from Argentinean plants was described [46] in which camphor (5.67%) was only the fourth most abundant compound, following 1,8-cineole (46.9%), 3-carene (19.81%) and α-pinene (7.43%). These authors distinguished toxicity by immersion, reporting an LC_50_ of 0.13 (95% CI = 0.019–0.49) mg/mL after 10 min of exposure, and by contact, with LC_50_ values of 15.2 (6–34.7) and 23.7 (12.1–69.6) mg/mL at 24 h for apterous and alates adults of *Metopolophium dirhodum*, respectively. A pure rosemary EO from EcoSMART Technologies Inc. (Franklin, TN, USA), predominantly composed of 1,8-cineole (31.5%), camphor (20.0%) and α-pinene (17.5%) yielded an LC_50_ of 10 mL/L (6.95–13.11) for adult female spider mites (*Tetranychus urticae*) reared on bean plants, and a slightly higher (13.0 mL/L, 95% CI = 10.05–17.78) for mites reared on tomato plants [47]. Another pure rosemary EO (Range Products Pty. Ltd., Perth, Australia), primarily composed of 1,8-cineole (35.27%), α-pinene (15.87%) and camphor (10.4%), resulted in 60% of aphid mortality at 5 μL/mL at 24 h [48], closely aligning with the results of the present study. Collectively, these findings suggest that camphor, 1,8-cineole, and the less frequently occurring cyclofenchene may be the key compounds responsible for insecticidal activity of rosemary EO.

*Laurus nobilis* L. (Lauraceae) is a native Mediterranean plant widely distributed across North Africa, Western Asia, and Southern Europe [49]. The EO extracted from its leaves was predominantly composed of monoterpene 1,8-cineole, followed by phenylpropanoid methyleugenol and monoterpene α-terpinyl acetate (Table 2). This laurel EO also achieved aphid mortality exceeding 80% (Table 3), reinforcing the potential significance of 1,8-cineole in insecticidal activity. Analysis of *L*. *nobilis* leaf EO collected in Southern Italy identified 55 compounds, comprising 91.6% of the total EO [50]. The principal components were 1,8-cineole (31.9%), sabinene (12.2%), and linalool (10.2%). Although these authors did not assess insecticidal activity, they reported antimicrobial and antifungal properties for the whole EO and isolated 1,8-cineole. Ben Jemâa et al. [51] studied the composition of EO from *L. nobilis* leaves collected from three African locations: Tunisia, Algeria, and Morocco. The three oils exhibited quantitative rather than qualitative differences in their chemical composition. 1,8-Cineole, linalool, and isovaleraldehyde were identified as the major common compounds, while α-pinene, α-terpineol, eugenyl methyl ether, β-pinene, spathulenol and β-myrcene were also well-represented in all three oils. The authors reported significant pest-repellent activity against *Tribolium castaneum* and *Rhyzopertha dominica* adults, concluding that insecticidal properties depended on the oils’ geographical origin, as the major EO components determine their biological properties. A recent report on the phytochemical constituents of *L. nobilis* fresh leaves EO revealed 1,8-cineole as the major compound (48.5%), followed by terpinyl acetate (13.5%), sabinene (9.2%) and α-pinene (4%) [52]. The authors also demonstrated that laurel EO exhibited broad-spectrum antimicrobial activity against all tested bacterial and *Candida* strains and significantly inhibited the growth of MCF-7 cancerous cells more effectively than the chemotherapeutic drug Doxorubicin. Furtado et al. [53] studied the chemical composition and biological activities of *Laurus azorica* EOs from four Azorean islands and *L*. *novocanariensis* from Madeira. All laurel EO samples were dominated by monoterpenes (76–97%), with oxygen-containing monoterpenes (OCMs) comprising 32–87% of the total EO. Regarding individual compounds, 1,8-cineole was identified as the major compound in three of the five EO analysed, while α-pinene was the predominant component in the other two. All laurel EOs and some of their major components applied individually exhibited strong repellent activity against medfly oviposition, as well as a moderate degree of mortality, which varied according to the plant EO’s geographical origin.

Regarding cypress EO, *Cupressus sempervirens* (commonly known as Italian or Mediterranean cypress) is found in subtropical Asia, North America, and the eastern Mediterranean. Pharmacological investigations have highlighted its diverse biological properties, including aromatherapeutic, antiseptic, astringent, balsamic or anti-inflammatory, astringent, diuretic, and antispasmodic activities [54]. The cypress EO analysed in this study was primarily composed of the monoterpenes cyclofenchene (21%) and 3-carene (15.56%), and the sesquiterpene cedrol (13,3%). Similar to the other EOs tested, the cypress EO achieved a maximum aphid mortality rate of 81% at the highest dosage (Table 3) despite its distinct chemical profile. Selim et al. [55] compared the chemical composition, antimicrobial and antibiofilm activity of the essential oil and methanolic extract of *Cupressus sempervirens* L. The cypress EO contained 20 identified constituents, representing 98.1% of the oil. The main components were α-pinene (48.6%), 3-carene (22.1%), limonene (4.6%) and α-terpinolene (4.5%), collectively comprising 79.8% of the oil. While the methanolic extract strongly inhibited the growth of the tested bacteria, the EO exhibited moderate antibacterial activity but no anti-*Candida* activity. A cypress EO composed primarily of cedrol (44.8%), followed by α-pinene (15.71%) and neoiso-dihydrocarveol acetate (9.7%) demonstrated maximum efficacy at 100 μL/L in fumigation toxicity bioassays and exhibited a high repellency rate (66.2%) [56]. Almadiy and Nenaah [57] identified α-pinene (49.1%), 3-carene (21.4%), and cedrol (5.1%) as the major components in a cypress EO from local gardens in southern Saudi Arabia. They reported LC_50_ values of 28.3 μg/mL and 12.6 μL/L for *Culex quinquefascitus* larvae and adults, respectively. Therefore, the insecticidal potential role of the EO in this species can be attributed to the monoterpenes cyclofenchene and 3-carene, and the sesquiterpene cedrol.

Although the rosemary, cypress, and laurel EOs, as characterised in this study, demonstrated comparable efficacy against aphids in contact toxicity bioassays, their potential as biopesticides requires further assessment regarding their impact on beneficial insects commonly used in Mediterranean greenhouses for biological control. The toxicity of these EOs against *Sphaerophoria rueppellii* larvae, one of the most effective natural predators of aphids in greenhouses, was found to be very low and largely attributable to the acetone solvent (Table 5). However, this is not always the case, as previous studies have shown that the coccinellids *Adalia bipunctata* and *Coccinella septempunctata* are 2 to 5 times more susceptible to EOs than aphids [58]. These findings underscore the absolute necessity of evaluating the effects of EO on beneficial fauna, given the varying sensitivities of different predators to such treatments. This necessity is especially critical when considering the application of EOs in integrated pest management, where the use of natural enemies (biological control) can be complementary to EOs (chemical control).

EOs from members of the Lamiaceae, Cupressaceae, and Lauraceae families obtained by hydrodistillation were compared, observing similar levels of toxicity against aphids and syrphid larvae. However, identifying a common factor that could explain the similar insecticidal potency of these three species is challenging. A comparison of the 10 most abundant compounds in the three EOs reveals no specific common compound. Although 1,8-cineole is present in both rosemary and laurel EOs, it is absent in cypress EO. This complexity in understanding the mechanisms by which EOs function as biopesticides is a major challenge that must be addressed to standardise these compounds for pest control applications. The only shared characteristic among the studied EOs is that their major fraction consists of monoterpenes, albeit with significant quantitative differences. Based on the distribution of monoterpenes, we hypothesise that the predominant presence of both bicyclic (e.g., camphor, borneol, β-pinene, 3-carene) and menthane-type (e.g., 1,8-cineole, α-terpinyl acetate, α-terpineol, terpinen-4-ol, terpinene) monoterpenes is crucial for the insecticidal activity of these EOs. However, this hypothesis requires validation through targeted experiments in the near future.

## 5. Conclusions

In this study, we evaluated three essential oils extracted by hydrodistillation from taxonomically distinct plant species: rosemary (*Lamiaceae*), laurel (*Lauraceae*), and cypress (*Cupressaceae*). When applied directly in acetone at a concentration of 10 μL/mL, these EOs exhibited high aphid mortality (exceeding 80%) while causing minimal mortality in their natural enemy, the syrphid *Sphaerophoria rueppellii*, in contact toxicity bioassays conducted in Petri dishes. These findings suggest that the tested EOs hold significant potential for aphid pest control. However, for effective implementation, it will be necessary to develop an optimised application method suitable for open-field conditions, which is currently under investigation.

## Figures and Tables

**Table 1 insects-16-00353-t001:** Summary of the composition and yield of rosemary, laurel, and cypress EO.

Compound Family	Rosemary	Laurel	Cypress
Carotenoids and apocarotenoids	0.05 ± 0.01 ^1^	-	-
Cyclic polyketides	0.01 ± 0.02	-	-
Fatty acyls	3.94 ± 0.02	0.82 ± 0.01	0.062 ± 0.001
Phenylpropanoids	0.788 ± 0.002	13.53 ± 0.14	0.177 ± 0.002
Monoterpenes	**82.31 ± 0.07** ^2^	**64.13 ± 0.14**	**58.03 ± 0.04**
Sesquiterpenes	11.90 ± 0.04	20.49 ± 0.40	37.00 ± 0.05
Diterpenes	0.087 ± 0.001	-	2.95 ± 0.11
Unidentified	0.20 ± 0.02	0.54 ± 0.13	0.26 ± 0.06
Others	0.72 ± 0.03	0.48 ± 0.01	1.52 ± 0.06
Total identified (%)	99.08 ± 0.05	98.97 ± 0.14	98.22 ± 0.12
Yield (mL EO 100 g dw^−1^)	1.18 ± 0.15	0.73 ± 0.03	0.43 ± 0.03

^1^ Sum of the relative abundances of the compounds belonging to each family. ^2^ Bold text was used to highlight the most abundant compound family in each EO. Data are the mean of two determinations ± standard deviation.

**Table 2 insects-16-00353-t002:** The ten more abundant compounds are found in rosemary, laurel, and cypress EO.

Plant	Compound	Compound Family	Relative Abundance (%) ^1^
Rosemary	Camphor	Monoterpenes	18.07 ± 0.02
	1,8-Cineole	Monoterpenes	13.95 ± 0.05
	Cyclofenchene	Monoterpenes	9.10 ± 0.30
	Borneol	Monoterpenes	6.28 ± 0.04
	Camphene	Monoterpenes	6.13 ± 0.01
	β-Pinene	Monoterpenes	4.14 ± 0.00
	Verbenone	Monoterpenes	4.09 ± 0.03
	3-Octanone	Fatty acyls	3.46 ± 0.01
	Sylvestrene	Monoterpenes	3.19 ± 0.03
	α-Terpineol	Monoterpenes	2.93 ± 0.03
Laurel	1,8-Cineole	Monoterpenes	17.13 ± 0.11
	Methyleugenol	Phenylpropanoids	11.34 ± 0.12
	α-Terpinyl acetate	Monoterpenes	10.88 ± 0.06
	Linalool	Monoterpenes	10.18 ± 0.01
	β-Phellandrene	Monoterpenes	5.04 ± 0.05
	3,6,6-Trimethyl-2-norpinene	Monoterpenes	3.40 ± 0.02
	α-Terpineol	Monoterpenes	2.81 ± 0.00
	β-Pinene	Monoterpenes	2.62 ± 0.03
	Terpinen-4-ol	Monoterpenes	2.58 ± 0.02
	Bicyclogermacrene	Monoterpenes	2.26 ± 0.03
Cypress	Cyclofenchene	Monoterpenes	21.06 ± 0.32
	3-Carene	Monoterpenes	15.56 ± 0.01
	Cedrol	Sesquiterpenes	13.31 ± 0.21
	Germacrene D	Sesquiterpenes	7.38 ± 0.01
	α-Terpinyl acetate	Monoterpenes	5.38 ± 0.61
	α-Terpinene	Monoterpenes	3.41 ± 0.01
	β-Funebrene	Sesquiterpenes	3.30 ± 0.00
	Sylvestrene	Monoterpenes	2.23 ± 0.00
	β-Myrcene	Monoterpenes	2.22 ± 0.00
	β-Cedrene	Sesquiterpenes	1.96 ± 0.00

^1^ Data are the mean of two determinations ± standard deviation.

**Table 3 insects-16-00353-t003:** Mortality of *Myzus persicae* in contact toxicity bioassays.

Essential Oil	Dose (μL EO/mL Acetone)	Mortality (%) Mean ± SD ^a^	LD_50_ (μL/mL) (95% CI) ^b^
Rosemary	2	34.0 ± 18.4	4.61 (2.69–7.89)
	4	40.0 ± 24.5	
	10	85.0 ± 21.7	
Laurel	2	42.0 ± 33.9	2.74 (1.32–5.59)
	4	76.0 ± 15.8	
	10	83.0 ± 11.6	
Cypress	2	52.0 ± 21.5	1.93 (0.70–5.28)
	4	77.0 ± 18.9	
	10	81.0 ± 12.9	
Control	-	6.0 ± 7.0	

^a^ Means with different letters indicate significant differences for each essential oil (*p* ≤ 0.05); ^b^ Lethal dose (LD_50_) (Confidence Interval (CI)).

**Table 4 insects-16-00353-t004:** Significance (*p*-values) obtained in Games-Howell post-hoc-test of aphid mortality data from Table 3.

		Treatment (Type of EO and Concentration)								
	**Control**	**R ^a^ 2 ^b^**	**R 4**	**R 10**	**L 2**	**L 4**	**L 10**	**C 2**	**C 4**	**C 10**
**Control**		**0.018**	**0.032**	**<0.001**	0.132	**<0.001**	**<0.001**	**0.001**	**<0.001**	**<0.001**
**R 2**	**0.018**		1	**<0.001**	0.999	**0.001**	**<0.001**	0.602	**0.002**	**<0.001**
**R 4**	**0.032**	1		**0.011**	1	**0.032**	**0.006**	0.969	**0.036**	**0.01**
**R 10**	**<0.001**	**<0.001**	**0.011**		0.084	0.982	1	0.07	0.995	1
**L 2**	0.132	0.999	1	0.084		0.208	0.074	0.998	0.208	0.1
**L 4**	**<0.001**	**0.001**	**0.032**	0.982	0.208		0.974	0.198	1	0.998
**L 10**	**<0.001**	**<0.001**	**0.006**	1	0.074	0.974		**0.03**	0.996	1
**C 2**	**0.001**	0.602	0.969	0.07	0.998	0.198	**0.03**		0.222	0.052
**C 4**	**<0.001**	**0.002**	**0.036**	0.995	0.208	1	0.996	0.222		1
**C 10**	**<0.001**	**<0.001**	**0.01**	1	0.1	0.998	1	0.052	1	

^a^ R: Rosemary EO; L: Laurel EO; C: Cypress EO; ^b^ 2: 2 μL/mL; 4: 4 μL/mL; 10: 10 μL/mL. Bold text was used to highlight the statistically significant differences (α = 0.05) between pairs.

**Table 5 insects-16-00353-t005:** Mortality of *Sphaerophoria rueppellii* larvae in contact toxicity bioassays.

Essential Oil	Dose (μL EO/mL Acetone)	Mortality (%) Mean ± SD
Rosemary	4	3.3 ± 12.9
	10	16.7 ± 24.4
Laurel	4	6.7 ± 17.6
	10	20.0 ± 25.4
Cypress	4	6.7 ± 17.6
	10	20.0 ± 31.6
Control 1 (acetone)		16.7 ± 30.9
Control 2		3.3 ± 12.9

## Data Availability

The original contributions presented in this study are included in the article/Appendix A; further inquiries can be directed at the corresponding author.

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
