# Peer review of "Evaluation of Selected Plant Essential Oils for Aphid Pest Control in Integrated Pest Management"

_insects, 2025, doi:10.3390/insects16040353_

Round 1
Reviewer 1 Report
Comments and Suggestions for Authors
The authors of this paper sought to chemically characterize three plant essential oils (rosemary, laurel, and cypress), and test their efficacy of controlling aphid pests as well as potentially detrimental effects on predatory syrphid larvae. I thought the paper was well-written overall and the experimental design was relatively straightforward. However, I do have some concerns that I think should be addressed.
Firstly, the authors should explain why the essential oils were blended with acetone for the experiment. Acetone may have unintended detrimental effects on insects, depending on the formulation. It looks like you designated a control with acetone alone (Control 1) for the syrphid larvae tests, but you did not do this for the aphids – why? Please explain this. It seems that acetone did have some effect on the syrphid larvae, so would you also expect this with the aphids? Without an acetone control, I don’t see how we would know if it is the acetone or essential oils that is contributing most to aphid mortality.
The longevity of a syrphid larva and an aphid are quite different. How might this play into the results of the experiment? Is 24 hours enough time to see any effects on syrphid larvae? Should the experiment be extended?
Another more general comment is that I do not think the authors dedicated enough time to discussing practical applications of essential oils in pest management. Do you think this would be more or less effective in a field trial? Is there existing data testing this? What might be the half-life of essential oils in a field trial and what is the best method of application?
In the methods, I cannot find where your sample size and replication is indicated. Please include this.
Table 5: mention that this is aphid mortality in the description as this is not immediately clear. Please also clearly indicate your replication.
I think that the data, as presented in table format, is adequate. However, I think that if it were presented in the format of a figure, it would be more impactful and easier to interpret.
Author Response
Please, see the attachment

Reviewer 2 Report
Comments and Suggestions for Authors
The manuscript describes the effects of essential oils from 3 plant species on Myzus persicae and on the predator of aphids, the hoverfly Sphaerophoria rueppellii. The essential oils were also characterized by GC-MS.
The paper addressees an interesting topic and adequately describes methods. I recommend minor revisions according to the attached document.

Reviewer 3 Report
Comments and Suggestions for Authors
The hypothesis of this research work is very good. Using EO for the control of aphids. I have concerns about the title. I think author need to revise it. < !--StartFragment -->This paper explain the effect of EO on aphid and syrphids. Not explaining anything about of aphids by syrphids. So this is not integrated pest management of aphids. So I think there is need to revise the title of this paper.
Some issues I have found in analysis. SD vales are very high. How authors will defend this. Need to see analysis part very carefully,
For detail comments please see the attached file.
< !--EndFragment -->

Author Response
Please, see the attachment.

Round 2
Reviewer 1 Report
Comments and Suggestions for Authors
I think the authors have appropriately responded to my comments. I have no further comments.